# A broadband achromatic polarization-insensitive metalens consisting of anisotropic nanostructures

Wei Ting Chen[1], Alexander Y. Zhu[1], Jared Sisler[1,2], Zameer Bharwani[1,2] & Federico Capasso[1]

Metasurfaces have attracted widespread attention due to an increasing demand of compact and wearable optical devices. For many applications, polarization-insensitive metasurfaces are highly desirable, and appear to limit the choice of their constituent elements to isotropic nanostructures. This greatly restricts the number of geometric parameters available in design. Here, we demonstrate a polarization-insensitive metalens using otherwise anisotropic nanofins which offer additional control over the dispersion and phase of the output light. As a result, we can render a metalens achromatic and polarization-insensitive across nearly the entire visible spectrum from wavelength $\lambda = 460$ nm to 700 nm, while maintaining diffraction-limited performance. The metalens is comprised of just a single layer of $TiO_2$ nanofins and has a numerical aperture of 0.2 with a diameter of 26.4 μm. The generality of our polarization-insensitive design allows it to be implemented in a plethora of other metasurface devices with applications ranging from imaging to virtual/augmented reality.

---

[1] Harvard John A. Paulson School of Engineering and Applied Sciences, Harvard University, Cambridge, MA 02138, USA. [2] University of Waterloo, Waterloo, ON N2L 3G1, Canada. Correspondence and requests for materials should be addressed to W.T.C. (email: weitingchen@seas.harvard.edu) or to F.C. (email: capasso@seas.harvard.edu)

Metasurfaces comprising sub-wavelength spaced nanostructures at an interface provide the means to accurately control the properties of light, including phase, amplitude, and polarization[1–4]. This allows for the possibility of highly compact and efficient devices[5–13]. Amongst these devices, metalenses have attracted intense interest due to their applicability to both consumer (phone cameras, virtual/augmented reality headsets) and industry products (microscopy, lithography, sensors, and displays)[14–23]. Recent works have focused on developing the broadband achromatic focusing capabilities of metalenses in the visible spectrum[24,25]. However, these metalenses suffer from polarization sensitivity, i.e., they can only focus light of a given circular polarization. This challenge can be overcome by using symmetric cylindrical or square-shaped nanopillars in both the visible[26] and the near-infrared regions[27–29]. However, by doing so, we lose a degree of freedom in the design space due to the symmetry of these constituent structures.

Here, counterintuitively, we show that it is indeed possible to simultaneously achieve an achromatic metalens capable of focusing any incident polarization in the visible using anisotropic TiO$_2$ nanofins. This is a different solution compared to recent publications associated with spatial multiplexing and symmetry[30–32]. These anisotropic nanofins allow us to accurately and simultaneously implement the phase and its higher-order derivatives (i.e., group delay and group delay dispersion) with respect to frequency. We designed and fabricated a metalens with a numerical aperture (NA) of 0.2. The metalens exhibits a measured focal length shift of only 9% $\lambda = 460$–700 nm and has diffraction-limited focal spots across the entire range. The focusing efficiency of the metalens varies by only ~ 4% under various incident polarizations. To showcase the generality of our principle, we also demonstrate a polarization-insensitive metasurface with diffraction efficiency of about 92% at wavelength $\lambda = 530$ nm.

## Results

**Principle of polarization-insensitive and achromatic focusing.** To achromatically focus a broadband incident beam in a diffraction limited spot, a metalens must impart a spatially and frequency-dependent phase profile given by

$$\varphi(r, \omega) = -\frac{\omega}{c}\left(\sqrt{r^2 + F^2} - F\right), \qquad (1)$$

where $r$, $\omega$, and $F$ are the lens radial coordinate, angular frequency, and a constant focal length, respectively. The Taylor expansion of Eq. 1:

$$\varphi(r, \omega) = \varphi(r, \omega_d) + \frac{\partial \varphi}{\partial \omega}\bigg|_{\omega=\omega_d}(\omega - \omega_d) + \frac{\partial^2 \varphi}{2\partial \omega^2}\bigg|_{\omega=\omega_d}(\omega - \omega_d)^2 + \dots \qquad (2)$$

identifies the required phase $\varphi(r, \omega_d)$, group delay $\frac{\partial \varphi}{\partial \omega}\big|_{\omega=\omega_d}$, and group delay dispersion $\frac{\partial^2 \varphi}{\partial \omega^2}\big|_{\omega=\omega_d}$ that needs to be fulfilled at every lens coordinate $r$. An intuitive way to understand each term in Eq. 2 is to treat the incident light as wavepackets. The required phase profile sends incident wavepackets towards the focal point, while the first and the higher order derivative terms ensure that the incident wavepackets arrive at the focal point simultaneously and identically in the time domain, respectively[24]. The challenge here lies in the fact that the chosen nanostructures must satisfy each derivative term in Eq. 2 at every lens coordinate. Previous designs made use of the geometric (or Pancharatnam-Berry) phase principle to decouple the phase, $\varphi(r, \omega_d)$, from the

dispersion (group delay and group delay dispersion)[24,25,33]. However, this approach also comes with an unwanted polarization-sensitivity, i.e. these achromatic metalenses can only focus incident light with a particular circular polarization.

Our design principle still involves Pancharatnam–Berry phase; however, we circumvent the aforementioned drawback by limiting the rotation angle of each anisotropic element to either 0 or 90 degrees. Each element is comprised of multiple nanofins to provide additional degrees of freedom to engineer the dispersion (Fig. 1a, inset). The layout of a quarter of our achromatic and polarization-insensitive metalens is depicted in Fig. 1a and a scanning electron microscope image from a region of our fabricated metalens is shown in Fig. 1b. To tune the phase and dispersion, each nanofin's length and width is varied and the gap (g) between nanofins is set to be either 60 nm or 90 nm. By using anisotropic elements instead of standard symmetric circular or square pillars[26,28], we have more geometric parameters to alter for better dispersion control. More importantly, the anisotropic elements offer the freedom to impart an additional $\pi$ phase shift without changing their dispersion characteristics. This is essential in order to fulfill both the required phase and dispersion given by Eq. 2, and can be understood from the Pancharatnam–Berry phase[34,35]. When light passes through a nanofin, the transmitted electric field can be described by the Jones vector:[36]

$$\begin{bmatrix} \tilde{E}_x \\ \tilde{E}_y \end{bmatrix} = \frac{\tilde{t}_l + \tilde{t}_s}{2}\begin{bmatrix} 1 \\ \pm i \end{bmatrix} + \frac{\tilde{t}_l - \tilde{t}_s}{2}\exp^{\pm i2\alpha}\begin{bmatrix} 1 \\ \mp i \end{bmatrix}, \qquad (3)$$

where $\tilde{t}_l$ and $\tilde{t}_s$ represent complex transmission coefficients when the normalized electric field of the incident light is polarized along the long and short axis of the nanofin, respectively. The $\alpha$ term is defined as the counterclockwise rotation angle of the nanofin with respect to the x-axis. The first term of Eq. 3 causes unwanted scattering and can be minimized if the nanofin is designed as a miniature half-waveplate. In this case, the amplitude of the second term $abs(\frac{\tilde{t}_l - \tilde{t}_s}{2})$ increases, corresponding to maximal polarization conversion efficiency. The $\exp^{\pm i2\alpha}$ in the second term is accompanied by a polarization converted term and illustrates the origin of Pancharatnam-Berry phase. Under left-handed circularly polarized incidence, a rotation of $\alpha$ imparts a frequency-independent phase of $2\alpha$ to the right-handed circularly polarized output light ($\begin{bmatrix} 1 \\ -i \end{bmatrix}$) without affecting the dispersion, which is determined by $\frac{\tilde{t}_l - \tilde{t}_s}{2}$. This usually results in polarization-sensitivity because the values of $\exp^{i2\alpha}$ and $\exp^{-i2\alpha}$, obtained under left and right circular polarized (LCP and RCP) incident light, respectively, are not identical. However, if one arranges the nanofin with $\alpha = 0°$ or $\alpha = 90°$, their values become equal. Therefore, both RCP and LCP incident light will experience the same phase profile upon interacting with a metalens consisting of either mutually parallel or perpendicular nanofins. Since any incident polarization can be decomposed into a combination of LCP and RCP, this property implies that the metalens is polarization insensitive, capable of focusing any incident polarization. Figure 1c confirms the results predicted by Eq. 3. A metalens element provides the same phase for both RCP (line) and LCP (circles) incidence, and, for a given circular polarization, a 90-degree rotation imparts a $\pi$ phase shift without affecting group delay (slope) and group delay dispersion (curvature).

**Design of an achromatic and polarization-insensitive metalens.** The design of our polarization-insensitive and achromatic metalens starts from a parameter sweep of the element shown in

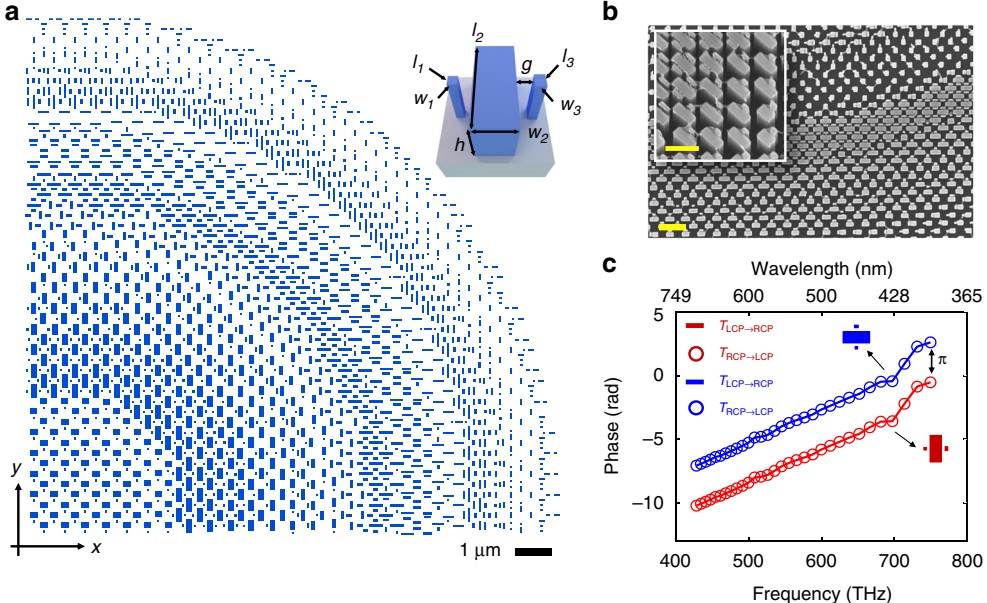

**Fig. 1** Principle behind polarization-insensitive and achromatic metalens. **a** Layout of a quadrant of the metalens. It has a NA of 0.2 and a diameter of 26.4 μm. The inset shows a schematic diagram of its constituent elements. Each element comprises $TiO_2$ nanofins with the same height $h = 600$ nm. These elements are spaced equally with a lattice constant of 400 nm. **b** A scanning electron microscope image of a part of the fabricated metalens. Scale bar: 1 μm. The inset shows a magnified and oblique view of the nanofins. Scale bar: 500 nm. **c** Simulated phase shift of the component of the transmitted electric field with polarization orthogonal to the incident circularly polarized light. The legend, for example $T_{LCP \to RCP}$, represents the phase of RCP transmitted light under LCP incidence. The blue and red colors show the same element, consisting of three nanofins, oriented along horizontal and vertical directions, respectively. The nanofin parameters ($w_1$, $l_1$, $w_2$, $l_2$, $w_3$, $l_3$, $g$) = (50, 50, 170, 370, 50, 90, 60) in nanometer units. The element shows identical phase under both RCP and LCP illuminations. Note that for a given incident circular polarization, a 90-degree rotation introduces a $\pi$ phase shift without affecting group delay (slope) and group delay dispersion (curvature)

the inset of Fig. 1a to build a library. We used a finite-difference time-domain (FDTD) solver to obtain each element's phase at $\lambda = 530$ nm, as well as its group delay and group delay dispersion. More simulation details can be found in our previous publication[24]. Figure 2a plots the three quantities of interest: phase, group delay, and group delay dispersion, at the design wavelength of 530 nm for each element. There are thousands of geometrical combinations, resulting in a dense scatter plot from which we identify the optimal elements to fine tune the dispersion. Note that due to the principle outlined in Fig. 1c, an element rotated by 90 degrees (i.e. purple points) will experience a $\pi$ phase shift for all frequencies with no change in the values of its dispersion. As a result, the design library can be further extended, allowing for better implementation of the required phase and dispersion (black symbols), which were calculated based on Eq. 1 for an achromatic metalens with a diameter of 26.4 μm and an NA of 0.2. To realize the metalens, the elements selected must be those closest to the required (black) points in the 3-dimensional space of phase, group delay, and group delay dispersion displayed in Fig. 2a. Because only the relative values of these parameters are important, the library can be shifted in this 3-dimensional space to better fit the required values. A particle swarm optimization method was used to find the optimal shifts for phase, group delay, and group delay dispersion, which minimizes the distance between each required point and the values provided by the elements in our library. The final results can be better visualized in Fig. 2b–d. The phase, group delay, and group delay dispersion of the selected metalens elements are shown in blue, together with the corresponding required values (black curves). We only consider terms up to the group delay dispersion because the values of any higher orders for our selected elements are very small.

**Focal spot and focusing efficiency characterizations**. We subsequently fabricated the achromatic and polarization insensitive metalens using electron beam lithography, followed by atomic layer deposition of $TiO_2$ and resist removal[37], and compared its performance to a chromatic metalens of the same diameter and NA. The chromatic metalens was designed using rotated nanofins with the same length and width to impart the Pancharatnam-Berry phase. The chromatic metalens represents the case without dispersion engineering and has a focal length shift similar to a Fresnel lens. We also show in Supplementary Movie 1 simulation results for a complete metalens with a smaller lens diameter and a higher NA of 0.6, confirming its achromatic and polarization-insensitive focusing behavior (Supplementary Figure 1). The focal length shifts of the fabricated achromatic and chromatic metalenses were determined by measuring their point spread functions at each wavelength along the propagation direction (z-axis) with 1 μm resolution (Fig. 3a). The left panel in Fig. 3a demonstrates a small focal length variation of about 6 μm for the achromatic metalens compared to that of 30 μm in the chromatic metalens (right panel). The normalized intensity profiles along the white dashed lines can be seen in Fig. 3b and Supplementary Figure 2 for the achromatic and chromatic metalenses, respectively. The achromatic metalens is diffraction-limited and its focal spot sizes and Strehl ratios as a function of wavelength are given in Supplementary Figure 3. Figure 3c shows achromatic imaging of a USAF resolution target from blue to red wavelengths in the visible. The results of imaging colored objects are given in Supplementary Figure 4. The achromatic metalens was also characterized by measuring the focusing efficiency of the focal spot under different polarizations of incident light. The focusing efficiency is defined as the focal spot power divided by transmitted

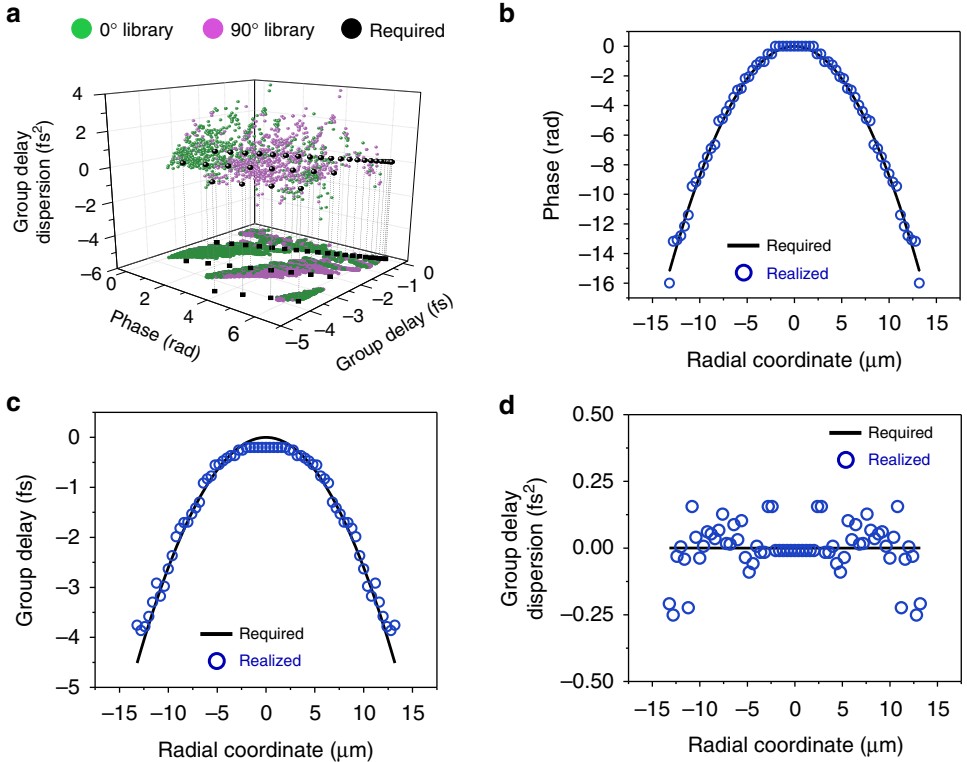

**Fig. 2** Required and realized phase and dispersion values for the metalens shown in Fig. 1a. **a** Phase, group delay, and group delay dispersion for all elements in our simulation library (colored points) and required values (black points). Each element (inset in Fig. 1a) is represented by a green and purple point in the plot because a 90-degree rotation can impart a phase change of $\pi$ without changing its group delay and group delay dispersion. **b–d** Realized (blue circles) and required (black curves) phase, group delay, and group delay dispersion at each radial coordinate across the polarization-insensitive and achromatic metalens

power through an aperture with the same diameter as the metalens. The measured focusing efficiencies weakly change with polarization, as shown in Fig. 3d. The inset shows focal spot profiles for different incident polarizations. These results experimentally prove that the metalens can focus any incident polarization. Note that the polarization state of the focal spot becomes the same as that of the incident light polarized along the axes of nanofins, which can be understood from the polarization converted term in Eq. 3. We attribute the efficiency variation to the interference of the focal spot with background light, i.e., the scattered light from the polarization conserved term (first term on the right-hand-side) in Eq. 3.

## Discussion

The diameter of the achromatic metalens is still small because it is limited by the achievable group delay in nanostructure elements[24]. The group delay is given as the height of the nanostructure divided by the group velocity of light; this height is limited due to fabrication constraints. Currently, we can achieve a group delay range of about 5 femto-second in our 600-nm-tall TiO$_2$ nanofins. There are some possible ways to circumvent this limitation, e.g. through hybrid diffractive-refractive lens design[38,39], high-aspect ratio nanofabrication to increase structure height[40,41] and using hyperbolic metamaterials to engineer group velocity over a large range[42–44].

It is worth noting that the metalens focusing efficiency shown in Fig. 3d is lower than our previous chromatic metalenses[19,45,46].

This can be explained by the fact that some elements with low polarization conversion efficiency were selected to cover a large range of dispersion values for achromaticity (see Supplementary Figure 5 for a plot of efficiency and dispersion). However, we emphasize that our approach does not preclude the design of highly efficient metasurfaces. For example, we show in Fig. 4a the layout of a conventional chromatic metasurface beam deflector designed for wavelength $\lambda = 530$ nm with an output diffraction angle of $\theta = 15°$. Figure 4b shows the normalized far-field power across the visible under $x$-polarized incidence as a function of wavelength. The metasurface has mainly a single diffraction order over a bandwidth of 50 nm centered at 530 nm and results in a high diffraction efficiency of about 92%. The diffraction efficiency is defined as the power of the first ($+1$) diffraction order divided by that of transmitted power. We numerically verified in Fig. 4c that such a high diffraction efficiency is maintained under various linearly and circularly polarized incident beams. It can be seen that at a given wavelength, the diffraction efficiency remains relatively constant across all polarizations, highlighting the polarization insensitivity of the metasurface. The absolute efficiency at $\lambda = 530$ nm, i.e. the power diffracted to 15 degrees divided by total incident power, is about 70% (see Supplementary Figure 6 for a plot of the absolute efficiency of the metasurface).

We have demonstrated with both simulations and experiments, a general principle for designing polarization-insensitive metasurfaces using anisotropic nanostructures as building blocks. These anisotropic structures allow for a more accurate

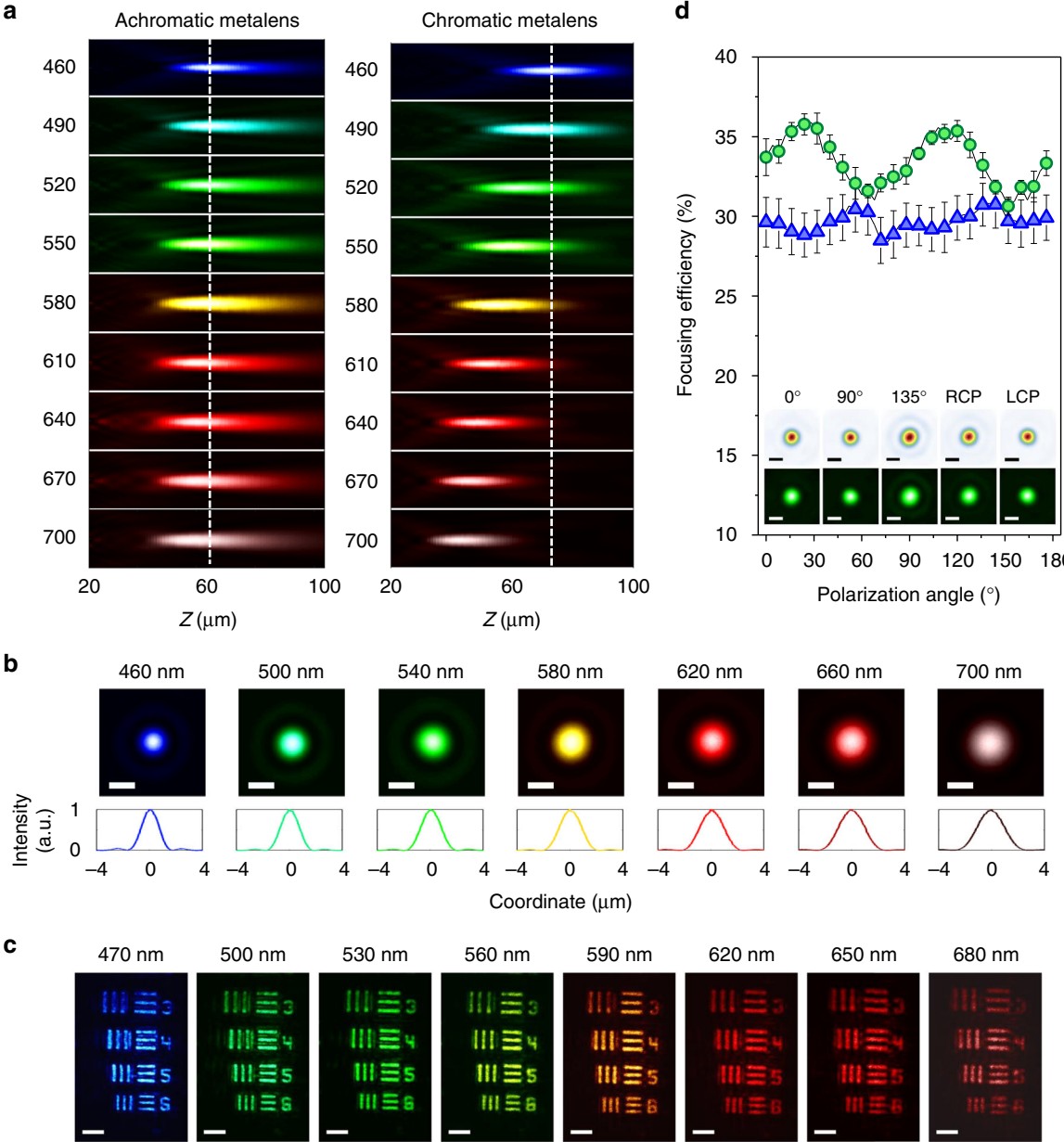

**Fig. 3** Measured focal spot profiles (in false colors), focusing efficiencies and imaging results. For focal spot profile and focusing efficiency measurements, the metalenses were designed with a NA = 0.2 and a focal length of 67 μm at λ = 560 nm. **a** Measured intensity distributions in the y-z plane shown in false colors corresponding to their respective wavelengths in the visible (labelled to left of plots in nanometers). The left and right panels correspond to achromatic and chromatic metalenses respectively. The latter, as a control sample, was designed without dispersion engineering and has a focal length shift similar to that of Fresnel lenses. Incident light travels along the positive z-axis. **b** Normalized intensity profiles along the white dashed lines of (a) for the achromatic metalens. The position of the dashed line corresponds to the focal length at λ = 460 nm. **c** Imaging with an achromatic metalens of NA = 0.05 and a diameter of 120 μm. The target is a standard USAF resolution chart. The pattern corresponding to number 6 has a linewidth of 8.77 μm. The light source is a tunable laser whose center wavelength is labelled on the top, and a bandwidth of 40 nm. The colors, brightness and contrast were adjusted for better visualization. A pair of polarization polarizer and analyzer was used to remove background light. Scale bars: 40 μm. **d** Focusing efficiency of the achromatic metalens (NA = 0.2) as a function of the angle of linearly polarized incident light in steps of 4°. The error bars span a range of two standard deviations. The illumination light sources are alternately a single wavelength 532 nm diode laser and a tunable broadband laser with 200 nm bandwidth centered at 570 nm. The measured focusing efficiencies using the monochromatic and broadband light source are represented by the green and blue symbols, respectively. The inset shows the focal spot profile, with the top and bottom rows corresponding to the diode (monochromatic) and tunable broadband laser illumination, respectively. The polarizations of input light are labelled at the top. Scale bars: 2 μm

implementation of phase, group delay, and group delay dispersion, while simultaneously making it possible to realize a polarization-insensitive, diffraction-limited and achromatic metalens from wavelength λ = 460–700 nm. Our design approach of polarization-insensitivity is also valid for other metasurface devices with applications in imaging and augmented reality.

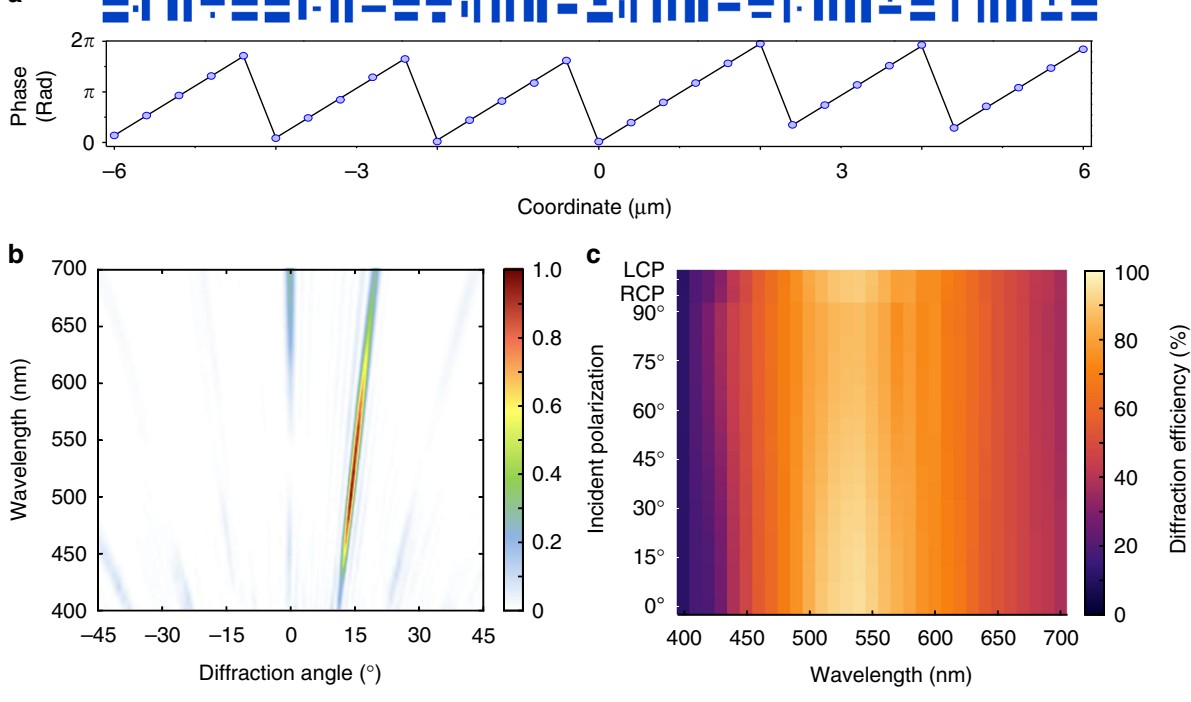

**Fig. 4** Simulated results for a polarization-insensitive phase-gradient metasurface. **a** Layout of the designed metasurface. The metasurface consists of mutually parallel and perpendicular nanofins with the geometries and orientations chosen to deflect a normal incident beam to an angle of 15 degrees at the design wavelength of 530 nm. The bottom panel shows the target and realized phases in a black line and blue circles, respectively. **b** Normalized far-field power under x-polarized incidence as a function of incident wavelength and diffraction angles. **c** Diffraction efficiency (colors) for the metasurface across the visible spectrum under linear and circular polarizations. The polarization angles are labelled on the y-axis, while the last two rows showing the cases for right- and left-handed polarizations. For all wavelengths, the efficiency is maintained at a relatively constant value, which is indicative of polarization insensitivity

## Data availability
The data that support the findings of this study are available from the corresponding author upon reasonable request.

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

## Acknowledgements

This work was supported by the Air Force Office of Scientific Research (MURI, grant# FA9550-14-1-0389 and grant# FA9550-16-1-0156) and the Defense Advanced Research Projects Agency (grant# HR00111810001). This work was performed in part at the Center for Nanoscale Systems (CNS), a member of the National Nanotechnology Coordinated Infrastructure Network (NNCI), which is supported by the National Science Foundation under NSF award no. 1541959. Federico Capasso gratefully acknowledges a gift from Huawei Inc. under its HIRP FLAGSHIP program.

## Author contributions

W.T.C. and F.C. conceived the study. A.Y.Z. fabricated the samples. W.T.C., J.S. and Z.B. performed simulations and developed codes. W.T.C., A.Y.Z. and J.S. measured the metalenses. All authors wrote the manuscript, discussed the results, and commented on the manuscript.

## Additional information

**Competing interests:** The authors declare no competing interests.

