## [Peer Review File · Nature Communications]

Reviewers' Comments:

Reviewer #1:

Remarks to the Author:

Chen et al., proposed and demonstrated a polarization-insensitive meta-lens with broadband and achromatic properties in the entire visible wavelength range. They verified their thought by controlling otherwise anisotropic nanofins design in both simulation and experiment. The authors also claimed the diffraction efficiency of the meta-lens could be reached to around 92%, and absolute efficiency is about 70%. The polarization-insensitive approach might be useful for potential applications on imaging. This manuscript has several important issues needs to be addressed before considerations for publication.

1. The authors emphasize that, at the wavelength of 530nm, diffraction efficiency is about 92% and absolute efficiency is around 70%. However, in Fig. 3, it shows the efficiency is only around 35%. What's the definition of the efficiency demonstrated in Fig.3? Note that the green and blue symbols show the measured efficiency at a single wavelength of 532nm and broadband light source in Fig. 3(c), respectively. Why the shape of the blue symbols looks like a sine function, while the variation of blue one looks quite small?

2. The author did not cite any recent works in the same field of polarization-insensitive metasurface. For example,

- (1) Arbabi, E, et al., 2016. Multiwavelength polarization-insensitive lenses based on dielectric metasurfaces with meta-molecules. *Optica*, 3(6), pp.628-633.
- (2) Ozer, A, et al., 2018. Polarization-insensitive beam splitters using all-dielectric phase gradient metasurfaces at visible wavelengths. *Optics letters*, 43(18), pp.4350-4353.
- (3) Sun, S, et al., 2017, May. All-dielectric metasurface for polarization-insensitive color printing. In *Lasers and Electro-Optics (CLEO), 2017 Conference on* (pp. 1-1). IEEE.
- (4) Lin, D, et al., 2018. Polarization-independent metasurface lens employing the Pancharatnam-Berry phase. *Optics Express*, 26(19), pp.24835-24842.
- (5) Shah, Y.D, et al., 2017. Ultra-narrow line width polarization-insensitive filter using a symmetry-breaking selective plasmonic metasurface. *ACS Photonics*, 5(2), pp.663-669.
- (6) Heidari, M.H. et al., 2018. Broadband wide-angle polarization-insensitive metasurface solar absorber. *JOSA A*, 35(4), pp.522-525.

These works should be cited and compared with the mechanism of this current work. And what's the advantage of the method in his work?

3. Line 157 on page 10, "It is worth noting that the meta-lens efficiency shown in Fig. 3(b) is lower ...". However, it seems that Fig. 3(b) shows intensity profiles for achromatic meta-lens and chromatic meta-lens, which makes readers confused.

4. The authors should check the format of reference carefully. For instance, there are some wired signs of Ref. 13, 14. Also there is no page number of Ref. 29.

Reviewer #2:

Remarks to the Author:

Broadband achromatic metasurfaces have attracted wide attention in the past year. The polarization-insensitivity is quite desired because the previous works all reported the circular polarization required metasurfaces. However, this manuscript entitled "A broadband achromatic polarization-insensitive metalens consisting of anisotropic nanostructures" is still lack for some important results that hinders it publishing in *Nature Communications*.

First of all, a lens needs to present an imaging result. Especially for such achromatic metalens, the

authors have to provide a full-color imaging by using their metalens sample. The achromatic focusing is not enough.

Secondly, the size of the fabricated metalens sample is too small, which might be the reason for lack for the imaging. The authors should give out a larger metalens sample. And as we know, there has a tradeoff between the size and NA value of an achromatic metalens. I hope that the authors discuss this point in the polarization-insensitive case and compare it with the polarization-sensitive metalens in their previous Nature Nanotechnology paper.

Last but not least, I wonder whether the nanostructures used in this work can cover all the phase, group delay and group delay dispersion case. Only after these nanostructures are able to present all the parameter case, this design approach can be useful for other achromatic metalenses and achromatic metasurfaces.

Reviewer #3:

Remarks to the Author:

Chen et al. experimentally demonstrate the achromatic and polarization-insensitive metalens at visible wavelengths by using TiO₂ meta-atoms. For a metalens with N.A. = 0.2, the measured focal length has a shift of only 9 % from wavelength $\lambda = 460$ nm to 700 nm. More interestingly, the focusing efficiency of the metalens varies by only ~ 4% under various polarizations of incident light. This is achieved by utilizing both the geometric P-B phase and propagation phase from the nanofin assembly. This is a very interesting and timing work, which provide useful route for designing achromatic and polarization independent metasurface lens. Therefore, I would recommend its publication in Nature Communications after addressing the following comments.

Comments:

1. The radius of the metalens is only tens of micron, I am wondering if it is easy to design a metalens with large area size, for example mm size metalens;
2. The numerical aperture of the metalens demonstrated in this work is 0.2, please specify if a larger N.A. could be designed and if yes it might be important to predict the optical efficiency;
3. Ref. 11, the page number is missing.

Reply to comments of Reviewer 1

Chen et al., proposed and demonstrated a polarization-insensitive meta-lens with broadband and achromatic properties in the entire visible wavelength range. They verified their thought by controlling otherwise anisotropic nanofins design in both simulation and experiment. The authors also claimed the diffraction efficiency of the meta-lens could be reached to around 92%, and absolute efficiency is about 70%. The polarization-insensitive approach might be useful for potential applications on imaging. This manuscript has several important issues needs to be addressed before considerations for publication.

1. The authors emphasize that, at the wavelength of 530 nm, diffraction efficiency is about 92% and absolute efficiency is around 70%. However, in Fig. 3, it shows the efficiency is only around 35%. What's the definition of the efficiency demonstrated in Fig.3? Note that the green and blue symbols show the measured efficiency at a single wavelength of 532nm and broadband light source in Fig. 3(c), respectively. Why the shape of the blue symbols looks like a sine function, while the variation of blue one looks quite small?

Response: The metalens efficiency is defined as the power contained in the focal spot, divided by the transmitted power through a circular aperture with the same diameter as the metalens. We have added this definition to our revised manuscript in the first paragraph on page 10 and changed the y-axis label in Fig. 3d to “focusing efficiency” to avoid confusion. The focusing efficiency of the lens is lower than the grating beam deflector because some low-efficiency elements were chosen to cover the large group delay region as required for achromatic focusing. This was mentioned in our original manuscript in the second paragraph on page 10.

The focusing efficiency variation for monochromatic laser illumination at 532 nm results from the interference of the focal spot and the background light, i.e. transmitted light which is not focused. This occurs because each fabricated nanostructure is not an ideal miniature half-waveplate, resulting in unwanted scattered light, which can be described by the first term of Eq. 3. It is less significant for broadband light because this variation is smoothed out due to the large bandwidth of the incident laser and the imperfection of achromatic waveplates used in our measurement. We have modified our manuscript to address this comment in the revised manuscript (first paragraph, page 10). In addition, we have simulated a small metalens and added the results to Fig. S1. Its focusing efficiency variation agrees well with our measurements.

2. The author did not cite any recent works in the same field of polarization-insensitive metasurface. For example,

(1) Arbabi, E, et al., 2016. Multiwavelength polarization-insensitive lenses based on dielectric metasurfaces with meta-molecules. *Optica*, 3(6), pp.628-633.

(2) Ozer, A, et al., 2018. Polarization-insensitive beam splitters using all-dielectric phase gradient metasurfaces at visible wavelengths. *Optics letters*, 43(18), pp.4350-4353.

(3) Sun, S, et al., 2017, May. All-dielectric metasurface for polarization-insensitive color printing. In *Lasers and Electro-Optics (CLEO), 2017 Conference on* (pp. 1-1). IEEE.

(4) Lin, D, et al., 2018. Polarization-independent metasurface lens employing the Pancharatnam-Berry phase. *Optics Express*, 26(19), pp.24835-24842.

(5) Shah, Y.D, et al., 2017. Ultra-narrow line width polarization-insensitive filter using a symmetry-breaking selective plasmonic metasurface. *ACS Photonics*, 5(2), pp.663-669.

(6) Heidari, M.H. et al., 2018. Broadband wide-angle polarization-insensitive metasurface solar absorber. *JOSA A*, 35(4), pp.522-525.

These works should be cited and compared with the mechanism of this current work. And what's the advantage of the method in his work?

Response: We have added these references into the revised paper, besides the last two papers which are related to plasmonics. We want to point out that these references didn't show any results related to achromatic metalenses. In our revised manuscript, we show not only the original achromatic focusing data but have also added achromatic imaging results. Our work presents a new approach to design achromatic and polarization-insensitive metalenses, which is advantageous because it allows us to control the dispersion (group delay and group delay dispersion) as well as phase profile of a given device; incidentally the above references only focus on manipulating the phase profile. These points are clearly mentioned in the introduction, as well as in the last paragraph of page 5.

3. Line 157 on page 10, "It is worth noting that the meta-lens efficiency shown in Fig. 3(b) is lower ...". However, it seems that Fig. 3(b) shows intensity profiles for achromatic meta-lens and chromatic meta-lens, which makes readers confused.

Response: We have corrected the typo. The sentence now reads "It is worth noting that the meta-lens focusing efficiency shown in Fig. 3(d)".

4. The authors should check the format of reference carefully. For instance, there are some wired signs of Ref. 13, 14. Also there is no page number of Ref. 29.

Response: We have fixed this problem, checked the reference format carefully and added page numbers to Ref. 29.

Reply to comments of Reviewer 2

Broadband achromatic metasurfaces have attracted wide attention in the past year. The polarization-insensitivity is quite desired because the previous works all reported the circular polarization required metasurfaces. However, this manuscript entitled "A broadband achromatic polarization-insensitive metalens consisting of anisotropic nanostructures" is still lack for some important results that hinders it publishing in Nature Communications.

First of all, a lens needs to present an imaging result. Especially for such achromatic metalens, the authors have to provide a full-color imaging by using their metalens sample. The achromatic focusing is not enough.

Response: We have fabricated another achromatic metalens following the same design approach, and used it for an imaging experiment. The achromatic metalens has a slightly smaller NA of 0.1. We have added the imaging results of a standard resolution target to Fig. 3(c) in the revised manuscript.

Secondly, the size of the fabricated metalens sample is too small, which might be the reason for lack for the imaging. The authors should give out a larger metalens sample. And as we know, there has a tradeoff between the size and NA value of an achromatic metalens. I hope that the authors discuss this point in the polarization-insensitive case and compare it with the polarization-sensitive metalens in their previous Nature Nanotechnology paper.

Response: To perform the imaging experiment you requested, we made the metalens a few times larger by lowering its numerical aperture to 0.1. We did so to increase the focal length of the achromatic metalens in order to prevent unintended physical contact between the metalens and the resolution target. The achromatic metalens may be still small but can provide good imaging quality as shown in Fig. 3c (The focusing results of the chromatic reference lens, originally in Fig 3b. have been moved to SI).

Compared to the metalens reported in our *Nature Nanotechnology* paper, the efficiency of our current metalens is doubled because it can focus any incident polarization. The diameter of this achromatic metalens is still small because the limitation of achievable group delay within current nanofabrication constraints remains. The *Nature Nanotechnology* paper has a detailed discussion of the trade-off between NA and metalens diameter. Indeed, there are several potential methods to overcome this limitation, such as hybrid diffractive-refractive design as discussed in the supplementary information of that paper. We have also discussed some of these possibilities in the second paragraph of page 10 and have cited related references.

Last but not least, I wonder whether the nanostructures used in this work can cover all the phase, group delay and group delay dispersion case. Only after these nanostructures are able to present all the parameter case, this design approach can be useful for other achromatic metalenses and achromatic metasurfaces.

Response: Figure 2 shows the library of the realized phase, group delay and group delay dispersion. Although there are some mismatches between the realized and required values due to a compromise between obtaining an ideal phase and dispersion profile, it is clear that the chosen structures can cover the target values.

Reply to comments of Reviewer 3

Chen et al. experimentally demonstrate the achromatic and polarization-insensitive metalens at visible wavelengths by using TiO₂ meta-atoms. For a metalens with N.A. = 0.2, the measured focal length has a shift of only 9 % from wavelength $\lambda = 460$ nm to 700 nm. More interestingly, the focusing efficiency of the metalens varies by only $\sim 4\%$ under various polarizations of incident light. This is achieved by utilizing both the geometric P-B phase and propagation phase from the nanofin assembly. This is a very interesting and timing work, which provide useful route for designing achromatic and polarization independent metasurface lens. Therefore, I would recommend its publication in Nature Communications after addressing the following comments.

Comments:

1. The radius of the metalens is only tens of micron, I am wondering if it is easy to design a metalens with large area size, for example mm size metalens.

Response: It is extremely challenging to realize a large diameter and continuously achromatic metalens across the entire visible spectrum in a single layer configuration. The reason is that the available group delay provided by a nanostructure is limited. This can be understood from the definition of group delay: nanostructure height divided by group velocity. The nanostructure height is limited due to fabrication constraints; currently we can achieve a group delay range of about 5 femto-seconds in our 600-nm-tall TiO₂ nanofins. However, as we have pointed out in the second paragraph on page 10 of revised manuscript, there are some approaches such as high-aspect ratio fabrication and hybrid diffractive-refractive design, which could potentially circumvent this challenge.

2. The numerical aperture of the metalens demonstrated in this work is 0.2, please specify if a larger N.A. could be designed and if yes it might be important to predict the optical efficiency.

Response: The NA can be made higher at the cost of lowering the diameter due to limitations in the range of achievable group delay, as discussed previously in our original manuscript in line 138 on page 9. In the original submission, we have included a supplementary movie showing a simulation of a full achromatic metalens. Its NA is 0.6, we have added more information regarding its efficiency in the supplementary information Fig. S1.

3. Ref. 11, the page number is missing.

Response: We have checked all reference formats carefully and have fixed this error.

Reviewers' Comments:

Reviewer #1:

Remarks to the Author:

I think the revised manuscript pretty much cleared out my questions in the last run.
Recommend for publication.

Reviewer #2:

Remarks to the Author:

I have carefully read the rebuttal of the manuscript entitled "A broadband achromatic polarization-insensitive metalens consisting of anisotropic nanostructures". Although the authors have provided their responses to my questions, the revised paper and their answers are not sufficient enough in my opinion.

As a broadband achromatic lens, a full-color imaging for objects with different colors should be demonstrated, such as that shown by Wang et al. *Nature Nanotechnology* 13(3), 227-232 (2018). However, the authors only present the imaging of resolution target under single color illumination, which is not enough to prove the broadband achromatic property of the metalens. The author should use their broadband achromatic metalens to imaging a colorful object and obtain a chromatic-aberration-free image that can confirm the achromatism of their metalens.

Most of all, recent publication of Shrestha et al. "Broadband achromatic dielectric metalenses," *Light: Science & Applications* (2018) 7:85 (DOI 10.1038/s41377-018-0078-x), shows an excellent and very similar results as that in this manuscript. I don't think this manuscript provide any better result and any new design approach than that of LSA paper. I cannot recommend the publication of this manuscript in *Nature Communication*.

Reviewer #3:

Remarks to the Author:

The authors have successfully replied my concerns on the size, NA and efficiency of the polarization insensitive achromatic metalens. Therefore, I support the publication of this work in *Nature Communications*.

Reply to comments of Reviewer 1

I think the revised manuscript pretty much cleared out my questions in the last run.
Recommend for publication.

Response: Thank you for your recommendation for publication.

Reviewer #2 (Remarks to the Author):

I have carefully read the rebuttal of the manuscript entitled "A broadband achromatic polarization-insensitive metalens consisting of anisotropic nanostructures". Although the authors have provided their responses to my questions, the revised paper and their answers are not sufficient enough in my opinion.

As a broadband achromatic lens, a full-color imaging for objects with different colors should be demonstrated, such as that shown by Wang et al. *Nature Nanotechnology* 13(3), 227-232 (2018). However, the authors only present the imaging of resolution target under single color illumination, which is not enough to prove the broadband achromatic property of the metalens. The author should use their broadband achromatic metalens to imaging a colorful object and obtain a chromatic-aberration-free image that can confirm the achromatism of their metalens.

Response: We have imaged colourful objects per your request. The results have been added in Supplementary Figure 4.

Most of all, recent publication of Shrestha et al. "Broadband achromatic dielectric metalenses," *Light: Science & Applications* (2018) 7:85 (DOI 10.1038/s41377-018-0078-x), shows an excellent and very similar results as that in this manuscript. I don't think this manuscript provide any better result and any new design approach than that of LSA paper. I cannot recommend the publication of this manuscript in *Nature Communication*.

Response: In the revised paper, we have cited this reference to replace its original conference proceedings report from the same group. Please be advised that the LSA paper reported achromatic metalens comprising cylindrical pillars in the near infrared, while ours is based on anisotropic nanofins in the visible.

Reviewer #3 (Remarks to the Author):

The authors have successfully replied my concerns on the size, NA and efficiency of the polarization insensitive achromatic metalens. Therefore, I support the publication of this work in *Nature Communications*.

Response: Thank you for your support for publication.